# Research on Dynamic Scheduling Model of Plant Protection UAV Based on Levy Simulated Annealing Algorithm

**Cong Chen, Yibai Li, Guangqiao Cao * and Jinlong Zhang ***

Nanjing Institute of Agricultural Mechanization, Ministry of Agriculture and Rural Affairs, Nanjing 210014, China
* Correspondence: caoguangqiao@126.com (G.C.); zhangjinlong@caas.cn (J.Z.)

**Abstract:** The plant protection unmanned aerial vehicle (UAV) scheduling model is of great significance to improve the operation income of UAV plant protection teams and ensure the quality of the operation. The simulated annealing algorithm (SA) is often used in the optimization solution of scheduling models, but the SA algorithm has the disadvantages of easily falling into local optimum and slow convergence speed. In addition, the current research on the UAV scheduling model for plant protection is mainly oriented to static scenarios. In the actual operation process, the UAV plant protection team often faces unexpected situations, such as new orders and changes in transfer path costs. The static model cannot adapt to such emergencies. In order to solve the above problems, this paper proposes to use the Levi distribution method to improve the simulated annealing algorithm, and it proposes a dynamic scheduling model driven by unexpected events, such as new orders and transfer path changes. Order sorting takes into account such factors as the UAV plant protection team's operating income, order time window, and job urgency, and prioritizes job orders. In the aspect of order allocation and solution, this paper proposes a Levy annealing algorithm (Levy-SA) to solve the scheduling strategy of plant protection UAVs in order to solve the problem that the traditional SA is easy to fall into local optimum and the convergence speed is slow. This paper takes the plant protection operation scenario of "one spray and three defenses" for wheat in Nanjing City, Jiangsu Province, as an example, to test the plant protection UAV scheduling model under the dynamic conditions of new orders and changes in transfer costs. The results show that the plant protection UAV dynamic scheduling model proposed in this paper can meet the needs of plant protection UAV scheduling operations in static and dynamic scenarios. Compared with SA and greedy best first search algorithm (GBFS), the proposed Levy-SA has better performance in static and dynamic programming scenarios. It has more advantages in terms of man-machine adjustment distance and total operation time. This research can provide a scientific basis for the dynamic scheduling and decision analysis of plant protection UAVs, and provide a reference for the development of an agricultural machinery intelligent scheduling system.

**Keywords:** plant protection UAV; dynamic scheduling; simulated annealing algorithm; Levy distribution

## 1. Introduction

Disease and pest control is an essential part of crop production and is related to food safety [1–3]. Plant protection unmanned aerial vehicles (UAVs) characterized by easy take-off and landing and high flight mobility are suitable for a variety of operating environments, such as plains and hills, and are, therefore, extensively applied in the process of crop disease and pest control [4–6]. In UAV plant protection operations, the swarm of plant protection UAVs generally provides services in the form of a UAV plant protection team, which can include multiple plant protection UAVs; multiple teams operate on plant protection orders by division of work and cooperation with each other. The reasonable scheduling of plant protection UAVs is crucial to guarantee the operation quality and improve the operation income of UAV plant protection teams [7]. Current studies on the scheduling of plant protection UAVs are mainly conducted from a static

perspective [8,9]. However, in practical operations, UAV plant protection teams often face unexpected situations, such as new orders and transfer cost changes, while the static model cannot adapt to the practical production demand properly. Therefore, a scheduling model of plant protection UAV swarms taking into account unexpected situations is of great practical significance to improve their operational efficiency and guarantee operational quality, while providing a reference for the development of scheduling systems for intelligent agricultural machinery [7].

Currently, the hot spot of research on the scheduling of plant protection UAVs focuses on in-field flight path planning, based on which the swarm scheduling model is solved by using the heuristic algorithm [8,9]. Li et al. [10] used the particle swarm algorithm to solve the UAV swarm allocation strategy based on in-field flight path planning of plant protection UAVs. Xu et al. [11] used the genetic algorithm to solve the operation sequence of plant protection UAVs based on UAV flight path planning in multiple fields. Cao et al. [12] sorted the operational plots by their size, distance, and operational urgency and optimized the scheduling path of plant protection UAVs by the elitist non-dominated sorting genetic algorithm (NSGA-II). Research on the scheduling of agricultural machinery similar to the scheduling scenario of plant protection UAVs started in the 1980s [13]. Wu et al. [14] proposed a multi-objective agricultural machinery scheduling model with time windows from the perspective of balancing operation income and quality and optimized the scheduling model by the dynamic planning method. Wang et al. [15] put forward a two-step scheduling model of clustering before allocation, considering the operational requirements of different crops, clustered the operational plots with such factors as attributes of planted crops and areas of operational plots as the measurement indexes, and optimized the planning model using the hybrid linear programming method on this basis. Gareth et al. [16] took the interface between rice harvesting and drying segments into comprehensive consideration and used the taboo search algorithm to optimize the scheduling model of rice harvesters with the goal of minimizing the interval between harvesting and drying.

To sum up, in terms of model optimization, the commonly used methods include two main categories: exact solution methods and heuristic algorithms [17,18]. Among them, exact solution methods include: dynamic programming, linear hybrid programming methods, etc. [19,20]. Heuristic methods include: particle swarm algorithm, tabu search algorithms, genetic algorithm, etc. [21–23]. The exact solution methods can be used to obtain the optimal solution of the planning strategy. However, they may face problems, such as dimensional explosion and long solution time under large-scale, multi-constraint conditions [24,25]. The heuristic algorithms have certain advantages in solving multi-constraint problems [25]. Among them, the simulated annealing algorithm (SA) is a commonly used heuristic algorithm [26]. However, the design of the computational function for accepting suboptimal solution probabilities during search makes the algorithm prone to falling into local optimum in the early stage and slow convergence in the later stage [27,28]. The results of previous studies have indicated that the Levi distribution has a good effect on preventing the heuristic algorithm from falling into local minimum [29,30].

In terms of scheduling model scenarios and optimization objectives, the optimization of plant protection UAV scheduling is a multi-objective optimization problem with time windows, and the optimization objectives include: maximizing operation income, minimizing operation time, and minimizing scheduling distance. The constraints of the optimization model include: the operation shall be conducted on each plot and completed within the operation time window. The above studies are mainly conducted from a static perspective, and the optimization models established lack the capacity to respond to unexpected situations in a timely manner [17–30]. However, in the practical operation process, the swarm of plant protection UAVs often faces dynamic events, such as additional orders and changes in the transfer cost of UAV plant protection teams due to COVID-19 or traffic congestion [31,32]. Therefore, it is more instructive to address the scheduling and planning issues of plant protection UAVs from a dynamic perspective, respond to unexpected sit-

uations in a timely manner, and establish a dynamic scheduling model that can adapt to changes in the environment for scheduling plant protection UAV swarms in practice.

According to the experiences in vehicle and military UAV scheduling, studies on dynamic scheduling models can be divided into two types: prediction of the occurrence probability of unexpected events and event-driven rescheduling [33–35]. Studies on the prediction of the occurrence probability of unexpected events include those from Amorim et al. [36] and Chang et al. [37]. Studies on event-driven methods include those form Wang et al. [38] and Grbac et al. [39]. Studies on the occurrence probability prediction of unexpected events mainly address the cases where the occurrence probability of an unexpected situation is predictable or the scheduling object planning is large, which can significantly increase the amount of computation in real-time scheduling. In the field of plant protection UAV scheduling, new orders or changes in road transfer costs are usually unpredictable factors, and the number of UAVs in a plant protection UAV swarm is limited, with relatively few optimization objectives. Hence, the event-driven dynamic scheduling method is more suitable for the scenario in this paper.

In summary, this paper fully considers the needs of farmers' plant protection operations, the actual operational capacity of UAV plant protection teams, the law of pest and disease spread, and other factors, and uses the methods of transportation logistics distribution scheduling and military equipment scheduling for reference to establish the operation scheduling models and algorithms of UAV plant protection teams in different operation scenarios. This will enable the provision of scientific and reasonable operation scheduling schemes for the UAV plant protection teams to meet their diversified scheduling needs and improve the utilization efficiency of UAV resources in order to ensure the economic benefits of UAV plant protection teams and reduce the losses caused by pests and diseases. The theoretical and practical significance of this study is reflected in the following aspects. (1) Upgrade the scheduling means of UAV plant protection teams, improve the informatization construction level of plant protection flight prevention teams, and enhance the response ability of flight prevention teams to farmer orders. (2) Improve the operational efficiency and economic benefits of the UAV plant protection teams, and reduce the risk of pest outbreak and spread within the region. (3) The dynamic scheduling method is introduced into the plant protection production scheduling problem to improve the response ability of the UAV plant protection teams to emergencies and provide a reference for the mechanical scheduling of other agricultural production links.

## 2. Problem Description

### 2.1. Description of Plant Protection UAV Scheduling Environment

In the practical operation links of plant protection, farmers issue plant protection operation orders to the UAV plant protection team according to the crop type and growth condition, and the order contents include: the location and area of the operation plot, operation time window and service type, etc. The UAV plant protection team organizes UAVs for the plant protection operation according to the workload of the received order. One team includes several UAVs, and the operation orders are completed by the division of work and cooperation among the teams.

Plant protection UAV scheduling is a multi-objective, multi-constraint optimization problem. Based on previous studies combined with practical operation demands [8–10], the optimization objectives of this paper are determined as follows: (1) minimum total operating time, (2) shortest total distance of UAV field scheduling, (3) highest total revenue of the UAV plant protection team, and (4) minimum penalty for delayed jobs. In the dynamic scenarios, mainly new orders and transfer cost changes are considered in this paper.

*2.2. Mathematical Description of the Model*

(1) Given that the description of the set of UAV plant protection teams: $U = \{U_1, U_2, U_3, \ldots, U_n\}$ denotes $n$ UAV plant protection teams, each can be described by Equation (1):

$$U_i = \{(l_i, t_i), e, v, C\} \tag{1}$$

In Equation (1), $l_i$ and $t_i$ denote the longitude and latitude of the current location of the UAV plant protection team, respectively; $e$ denotes the operation efficiency of the plant protection UAV; $v$ denotes the field transfer efficiency of the plant protection UAV; $C$ denotes the set of operation income and the cost of the plant protection UAV.

(2) Set of operation income and cost

$$C = \{C_s, C_{wo}, C_t, C_{wa}, C_{de}\} \tag{2}$$

The set $C$ mainly includes: the operation income ($C_s$) of the plant protection UAV; the operation cost ($C_{wo}$) of the plant protection UAV, such as energy, pesticide, and machinery depreciation; the transfer cost ($C_t$) of the plant protection UAV; the waiting cost ($C_{wa}$) of the plant protection UAV; and the penalty cost ($C_{de}$) of the plant protection UAV beyond the operation time window.

(3) Set of farmland orders

$$F = \{(l, t), (t_s, t_e), a, e\} \tag{3}$$

The set of farmland orders mainly contains: $(l,t)$: the longitude and latitude of the area where the farmland is located; $(t_s, t_e)$ denote the start and end time of the farmland operation time window, respectively; $a$: plot area; $e$: urgency of plot operation.

(4) Set of UAV plant protection team transfer costs ($TP = \{PF, PU\}$). The transfer costs of UAV plant protection teams include the transfer cost from each UAV plant protection team to each farmland and the path transfer cost between farmlands.

(5) Set of operation flags ($JS = \left\{x_f\right\}$). The operation flag has two values, indicating whether the operation is completed in each farmland, with 1 being operation completed, and 0 being operation not completed.

(6) Set of operation time ($WT = \{FW_s, FW_e\}$). $FW_s$ denotes the start time of the plant protection operation by the plant protection UAV on farmland $F$, and $FW_e$ denotes the end time of the plant protection operation on farmland $F$.

*2.3. Mathematical Model of Operation Scheduling*

2.3.1. Objective Function

(1) Maximize operation income

$$\max C = \sum_{i=1}^{m}(C_{si} - C_{woi} - C_{wai} - C_{ti} - C_{dei}) \tag{4}$$

$C_{si}$ denotes the operation income of the plant protection UAV on plot $i$, which is proportional to the area of the farmland and the urgency of the operation; $C_{woi}$ denotes the operation cost of the machinery and pesticides on plot $i$, which is proportional to the area of plot $i$; $C_{wai}$ denotes the cost of waiting for the plant protection UAV operation; $C_{ti}$ denotes the cost of transfer to plot $i$ in the plant protection UAV operation strategy; $C_{dei}$ denotes the corresponding penalty for the delayed operation of the plant protection UAV on plot $i$. Among them, $C_{ti}$, $C_{wai}$, and $C_{dei}$ can be reduced through the reasonable scheduling strategy to increase the total revenue of the UAV plant protection team.

(2) Minimize total operation time

$$\min T = \max(FW_e) - \min(FW_s) \tag{5}$$

The total operation time ($T$) can be expressed by Equation (5), where $FW_e$ denotes the operation end time of orders; $\max(FW_e)$ denotes the latest operation end time of all orders; $FW_s$ denotes the operation start time of orders; $\min(FW_s)$ denotes the earliest operation start time of all orders. The total operation time of the UAV plant protection team can be expressed as the difference between the latest operation end time and the earliest operation start time.

$$\max R = \sum_{i=1}^{m}(C_{si} - C_{woi} - C_{ti} - C_{wai} - C_{dei}) - (\max(FW_e) - \min(FW_s)) \tag{6}$$

In summary, the objective of operation scheduling can be described as Equation (6): $R$ denotes the scheduling synthesis target value, which consists of maximizing service income and minimizing operational time, where each part shall be normalized to solve the problem of different dimensions.

### 2.3.2. Constraint Function

The following constraints shall be met for the operation scheduling of UAV plant protection teams.

$$\forall x_f = 1 \tag{7}$$

$$WT \in [t_s, t_e] \tag{8}$$

Equation (7) indicates that operation services shall be completed on all farmlands. Equation (8) indicates that the operation time on each farmland shall be completed within the operation time window of farmland demand.

## 3. Design of Scheduling Model Based on Levy Distribution

The scheduling model of plant protection UAV mainly includes two parts: operation order sorting and order task assignment.

### 3.1. Prioritization Rules for Plant Protection Orders in Static Situations

The following factors are mainly considered for operation order sorting:

(1) Operation time window. When issuing orders, farmers can set the operation time window of a plot based on the comprehensive consideration of factors such as disease and pest outbreak patterns, crop growth, or local weather. The UAV plant protection team needs to carry out the plant protection operation within the operation time window. Excessively early or late operation will not be conducive to achieving the crop control effect; for operation beyond the time window, a certain penalty of reduced income will be imposed on the UAV plant protection team. Therefore, the earlier the time window of the plot starts, the shorter the total duration of the time window and the higher the order priority.

(2) Plot area. In terms of plant protection UAV operation efficiency and revenue, the larger the field plot area, the more regular the field shape, the higher the operation efficiency of the plant protection UAV, and the higher the revenue of the UAV plant protection team. Therefore, the larger the field plot area, the higher the order priority.

(3) Operational urgency. Farmers can add the operational urgency flag according to the urgency of the operation. For urgent operation orders, the UAV plant protection team can get a higher pay. Therefore, the higher the order urgency, the higher the order operation priority.

In this paper, the plant protection order sorting weights are calculated according to Equation (9).

$$o = w_1 \cdot t_s + w_2 \cdot (t_e - t_s) + w_3 \cdot a + w_4 \cdot e \tag{9}$$

$$w_1 + w_2 + w_3 + w_4 = 1 \tag{10}$$

In Equation (9), the start time of the time window is $t_s$, the duration of the time window is $(t_e - t_s)$, the operation plot area is $a$, and the priority is $e$. All the above variables are

subject to normalization. $w_1$ denotes the weight of the time window, $w_2$ denotes the weight of the operation time window duration, $w_3$ denotes the weight of the field plot area, $w_4$ denotes the urgency weight, and the sum of all weights is 1, as shown in Equation (10). After the calculation of operational priority weights for all plots, the operational sequence of all field plots is obtained by sorting in descending order.

*3.2. UAV Scheduling Solution Model Based on Simulated Annealing Algorithm with Levy Distribution*

Based on order sorting, tasks shall be assigned to UAVs in the plant protection team. The scheduling cost of plant protection UAVs is calculated according to Equation (6). The simulated annealing algorithm is characterized by easy process and fast search for multi-dimensional problems. Therefore, the scheduling strategy of the plant protection UAV is optimized by the simulated annealing algorithm in this paper based on the annealing principle of solid. When the solid temperature is very high, its internal energy is relatively large, and its internal particles are in rapid disorderly operation. When the temperature drops slowly, the internal energy of the solid decreases, and the motion of particles gradually becomes orderly. Finally, when the solid is at room temperature, the internal energy reaches the minimum. At this point, the particles are in the most stable state. The process of the simulated annealing algorithm is as follows:

(1) Let the initial temperature be T and the maximum number of cycles be *M*.
(2) Randomly generate a set of allocation strategies ($w$) and calculate the cost function score $f(w)$ under this strategy.
(3) Randomly generate a set of perturbation allocation strategies ($w'$) and calculate the cost function score $f(w')$ under this strategy.
(4) Calculate $\Delta E = f(w') - f(w)$.
(5) If $\Delta E < 0$, the perturbation strategy $w'$ is accepted; otherwise, the probability of accepting the perturbation strategy is calculated according to Equation (9), where T is the current temperature. That is, a number δ between (0,1) is randomly generated. If $\delta < p$, the perturbation strategy $w = w'$ is accepted; otherwise, the original strategy $w = w$ is accepted. The number of cycles is $i = i + 1$.

$$p = \frac{1}{1 + e^{-\Delta E/T}} \tag{11}$$

(6) If the strategy w meets the optimization requirement, or $i > M$, then the cycle is broken; otherwise, $T = T - t$, where $t$ is the value of the temperature drop in each cycle, and proceed to step (3).

The traditional simulated annealing algorithm is analyzed: The probability of accepting the suboptimal solution by the traditional simulated annealing method is shown in Equation (11). Equation (11) is a composite function of $1 + e^{-x}$ and $\Delta E/T$, where $1 + e^{-x}$ is a curve function, generally known as sigmoid function. The function curve is shown in Figure 1. It can be seen from Figure 1 that the function values are distributed in the interval [0,1], and the function changes fast in [0,1]. As the independent variable increases, the function value tends to 1 infinitely.

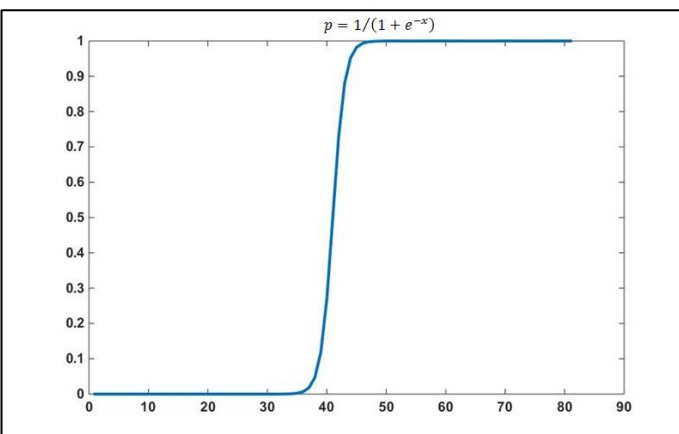

**Figure 1.** Schematic diagram of sigmoid function.

$\Delta E/T$ is a random function. According to the description of the simulated annealing algorithm process, $\Delta E$ is a random variable and $T$ is a linear variable. To describe in line with the function variation law, this paper assumes that $\Delta E$ is a random number that follows a Gaussian distribution, and linear variable T is described by Equations (12) and (13). The MATLAB platform is used to describe the composite function, and the simulated distributions of $p = \frac{1}{1+e^{-\Delta E/T}}$ are shown in Figure 2a,b. The results in Figure 2 indicate that the function value fluctuates substantially in the early stage of iteration, which is not conducive to the exchange of suboptimal and optimal solutions, making it difficult to achieve the purpose of expanding the search range and jumping out of the local optimum solution. In the later stage of iteration, most of the function values are distributed around 0.5. As a result, there is still a 0.5 probability of accepting the suboptimal solution based on the method, which is not conducive to the convergence of function values.

$$T = 1000 - 8x \tag{12}$$

$$T = 100 - 8x \tag{13}$$

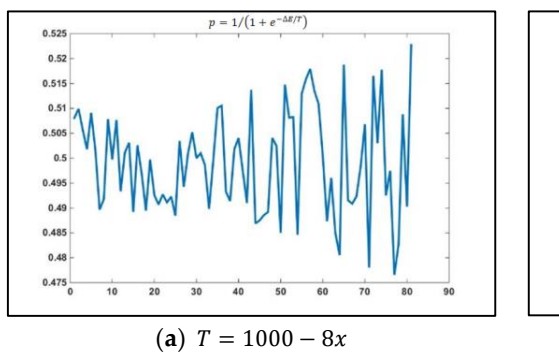 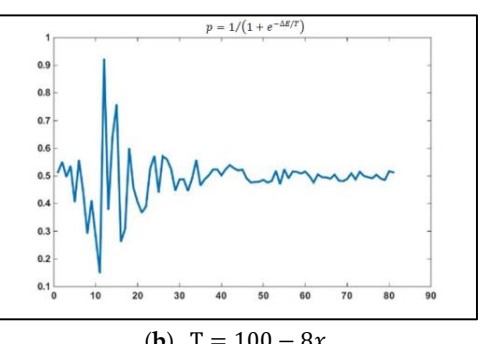

(**a**) $T = 1000 - 8x$     (**b**) $T = 100 - 8x$

**Figure 2.** Schematic diagram of $p = \frac{1}{1+e^{-\Delta E/T}}$ function.

As the improved simulated annealing algorithm has the problems of weak search ability in the early stage and slow convergence in the later stage, this paper proposes to improve the probability function for accepting suboptimal solutions in the simulated annealing algorithm by using the Levy distribution method. The Levy distribution can be expressed by Equation (14), where Levy distributions vary with different values taken for $C$. Figure 3 shows the function distribution corresponding to different values taken for the constant $C$. The figure indicates that regardless of the value taken for $C$, the range of function values of Levy distribution is [0,1]; as the independent variable increases, the

function values increase rapidly, then decline slowly and level off, and finally stabilize at around 0.1. With the increasing value of *C*, the maximum of the function declines.

$$f(x;\mu,c) = \begin{cases} 0, \mu < 0 \\ \sqrt{\dfrac{c}{2\pi}}\dfrac{e^{-\frac{c}{2(x-\mu)}}}{(x-\mu)^{\frac{3}{2}}}, \mu \geq 0 \end{cases} \tag{14}$$

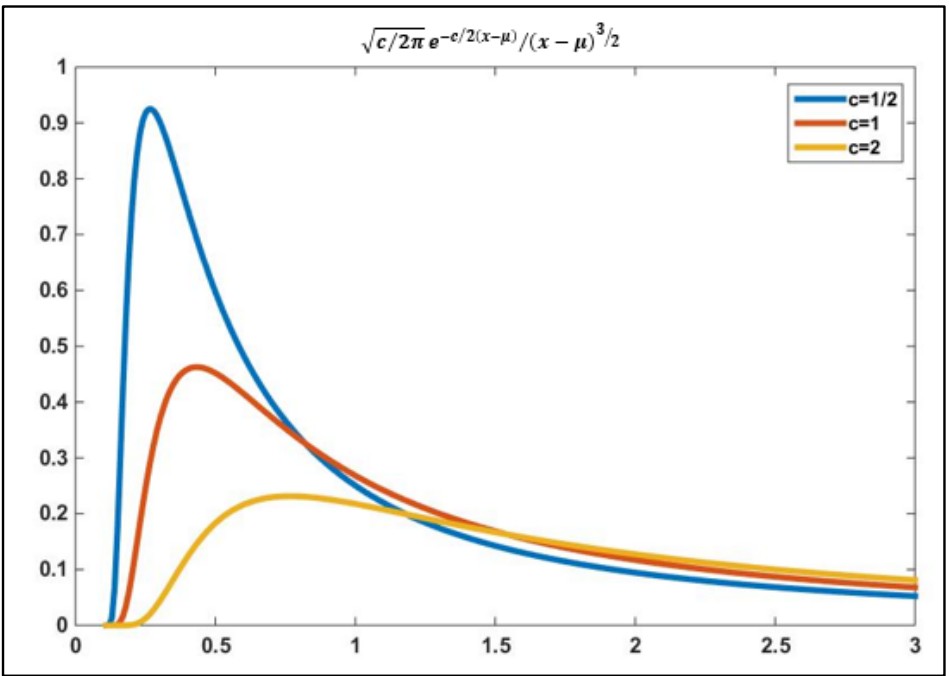

**Figure 3.** Levy distribution.

The function that accepts suboptimal solutions based on the simulated annealing algorithm is improved by the Levy distribution method, and the function can be re-expressed as Equation (15), where T is replaced by Equations (12) and (13). The composite function is shown in Figure 4a,b. Figure 4 shows that no matter how the T-value function changes, the composite function still retains the function trend of the Levy distribution; that is, as the independent variable gradually increases, the function value increases first, then decreases, and gradually stabilizes at around 0. By applying the composite Levy distribution to the simulated annealing algorithm, the following can always be achieved: At the beginning of the iteration, the temperature T is relatively high, the probability function value is high, and the function has a relatively high probability of accepting new solutions, which is conducive to jumping out of the local minimum and obtaining the global optimum. As the number of iterations increases, the temperature T keeps dropping, the iteration region converges, and the probability of accepting new solutions gradually decreases, accelerating the convergence of the function.

$$f(x;\mu,c) = \begin{cases} 0, \Delta E < 0 \\ \sqrt{\dfrac{T}{2\pi}}\dfrac{e^{-\frac{T}{2(\Delta E)}}}{(\Delta E)^{3/2}}, \Delta E \geq 0 \end{cases} \tag{15}$$

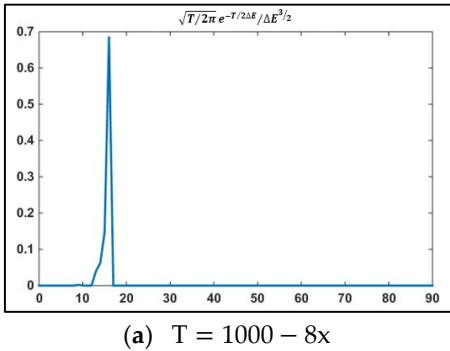 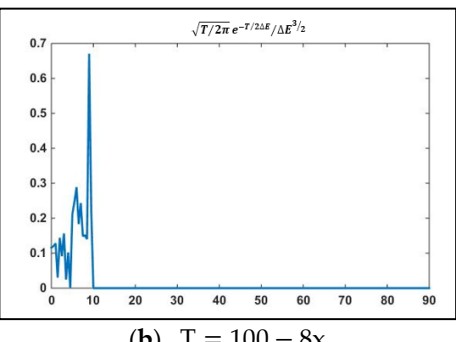

(**a**)  T = 1000 − 8x          (**b**)  T = 100 − 8x

**Figure 4.** Schematic diagram of $\sqrt{\dfrac{T}{2\pi}}\dfrac{e^{-\frac{T}{2(\Delta E)}}}{(\Delta E)^{3/2}}$.

.

The probability function in step (5) of the simulated annealing algorithm is improved by using the Levy distribution. The steps of the improved Levy simulated annealing algorithm (Levy-SA) are as follows:

(1)  Let the initial temperature be $T$ and the maximum number of cycles be $M$.

(2)  Randomly generate a set of allocation strategies ($w$) and calculate the cost function score $f(w)$ under this strategy.

(3)  Randomly generate a set of perturbation allocation strategies ($w'$) and calculate the cost function score $f(w')$ under this strategy.

(4)  Calculate $\Delta E = f(w') - f(w)$.

(5)  If $\Delta E < 0$, the perturbation strategy $w'$ is accepted; otherwise, the probability of accepting the perturbation strategy is calculated according to Equation (9), where T is the current temperature. That is, a number δ between (0,1) is randomly generated. If $δ < p$, the perturbation strategy $w = w'$ is accepted; otherwise, the original strategy $w = w$ is accepted. The number of cycles is $i = i + 1$.

(6)  If the strategy $w$ meets the optimization requirement, or $i > M$, then the cycle is broken; otherwise, $T = T - t$, where $t$ is the value of the temperature drop in each cycle, and proceed to step (3).

## 4. Design of Dynamic Scheduling Model

In the actual process of the plant protection operation, there may be a variety of unexpected emergencies, such as weather changes in a place will lead to a shorter working time, resulting in farmers in the area calling the plant protection operation in advance; if a major traffic accident occurs in the operation area, the UAV plant protection teams must take a detour. Therefore, in the process of carrying out the original operation plan, it is not rigid to carry out the operation step by step with the fixed operation plan. It is necessary to take the original operation plan and emergencies into account and reschedule the flight prevention task.

In the dynamic model planning, mainly the dynamic scenarios of new orders and path transfer cost changes of the plant protection UAV are considered, which occur frequently in the practical operation of UAV plant protection teams. Inspired by references [21–23], this paper transforms the dynamic scheduling process into an event-driven static scheduling model to obtain the solution. When the dynamic event occurs, the solution of the scheduling model is obtained again to achieve the goal of maximizing the operation income and minimizing the operation time while meeting the operation time window requirement. Specific scheduling ideas are shown in Figure 5.

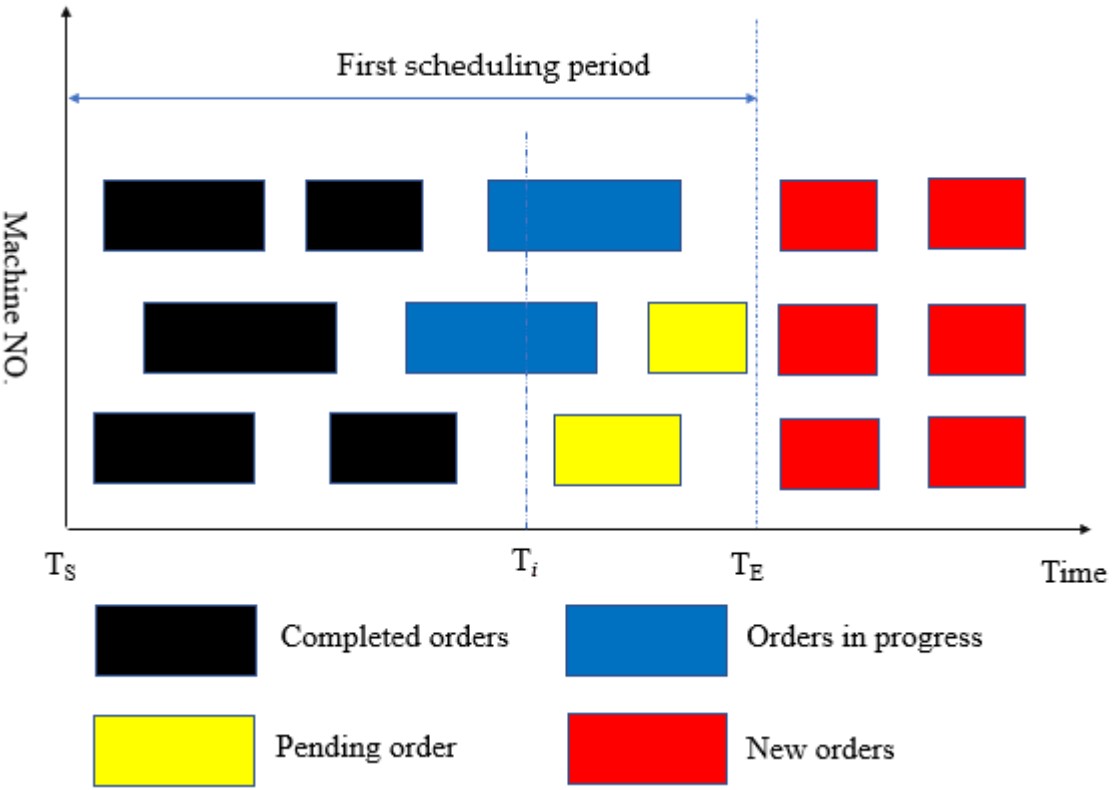

**Figure 5.** Dynamic scheduling strategy.

$T_s$ and $T_e$ are the start and end times of the first scheduling, respectively. If there are no unexpected circumstances, all plant protection operations originally planned will end at $T_e$. However, at $T_i$ time before the end of the operation plan, some new orders or some road traffic changes must be made to the original operation plan. In the event of an emergency, all operation orders can be classified into four categories. The first type is completed orders, i.e., operation orders that have been completed before the emergency occurs. The second type is orders in progress, i.e., orders that have been scheduled in the first dispatch, but have not yet finished when an emergency occurs. The third type is pending orders, i.e., orders that have been scheduled in the first dispatch, but have not yet started when an emergency occurs. The fourth type is new orders, i.e., orders added in the event of an emergency without a scheduled operation. Since the first and second types of orders have been closed or started, they cannot participate in repeat scheduling. The orders involved in rescheduling are those of the third and fourth types that have not yet started. The dynamic scheduling model is designed as follows, according to new orders and traffic changes.

### 4.1. Dynamic Scheduling Model in the Scenario of New Orders

Where there are new orders during the operation of the UAV plant protection teams, this paper proposes the following rules for managing the orders and scheduling the plant protection UAVs:

(1) Management of original orders. Those in the original orders that have not been operated upon by the arrival of new orders are $NF = \{F \mid notdo(F)\}$, and the plant protection UAV in operation is $NU = \{U \mid doing(U)\}$.

(2) Sorting of operation orders. The new order $NE$ is merged with the unoperated order $NF$ to form a new order set $FD = \{NE \cup NF\}$, based on which the new order set $FD$ is sorted by the method in Section 3.1.

(3) Management of available plant protection UAVs. The difference set between the set of all UAVs ($U$) and the set of plant protection UAVs that are operating ($NU$) is obtained to identify the set of available plant protection UAVs ($UD = \{U - NU\}$). The location

of each plant protection UAV in the set *UD* is updated. The path transfer cost from each plant protection UAV in the set *UD* to each plot in the *NU* is updated.

(4)  Assignment of tasks. The Levi simulated annealing algorithm is used to calculate the UAV task assignment.

*4.2. Dynamic Scheduling Model in the Scenario of Transfer Path Change*

Where the transfer path is changed due to COVID-19, road construction, and other unexpected situations, this paper proposes the following rules for managing the plant protection orders and scheduling the plant protection UAVs:

(1)  Management of original orders. Those in the original orders that have not been operated upon by the change of path cost are *NF* = {*F* | notdo(*F*)}, and the plant protection UAV in operation is *NU* = {*U* | doing(*U*)}.

(2)  Management of available plant protection UAVs. The difference set between the set of all UAVs (*U*) and the set of plant protection UAVs that are operating (*NU*) is obtained to identify the set of available plant protection UAVs (*UD* = {*U* − *NU*}). The location of each plant protection UAV in the set *UD* is updated. The path transfer cost from each plant protection UAV in the set *UD* to each plot in the *NU* is updated.

(3)  Assignment of tasks. The Levi simulated annealing algorithm is used to calculate the UAV task assignment.

## 5. Experimental Scenario and Environment

*5.1. Description of Experimental Object and Scenario*

In this paper, the plant protection flight operations of "one spraying and three prevention" on wheat in the Jiangning and Luhe districts of Nanjing in mid to late April are taken as the study object. It is assumed that five UAV plant protection teams were involved in the plant protection tasks in the aforesaid. The initial location of the teams and the location, area, and time window of the field plots are shown in Tables 1 and 2. The earliest operation order started on 11 April, the duration of the operation time window was 4–7 days, and the area of each operation plot was in the range of 100–250 hm$^2$.

**Table 1.** Initial position information of UAV plant protection team and UAV.

| Number | Longitude | Latitude | Number of UAVs |
|--------|-----------|----------|----------------|
| 1 | 118°22′ E | 31°14′ N | 8 |
| 2 | 119°08′ E | 32°19′ N | 6 |
| 3 | 119°45′ E | 32°20′ N | 6 |
| 4 | 119°08′ E | 31°04′ N | 10 |
| 5 | 120°12′ E | 32°29′ N | 4 |

**Table 2.** Original order information.

| Number | Longitude | Latitude | Area/hm$^2$ | Operation Time Window | Urgency |
|--------|-----------|----------|-------------|------------------------|---------|
| 1 | 118°22′ E | 31.14′ N | 124.53 | 11 14 | 1 |
| 2 | 118°54′ E | 31.12′ N | 130.31 | 11 15 | 0 |
| 3 | 118°89′ E | 31.28′ N | 225.54 | 12 16 | 0 |
| 4 | 118°32′ E | 31.26′ N | 138.23 | 13 17 | 1 |
| 5 | 118°35′ E | 31.25′ N | 148.98 | 14 17 | 0 |
| 6 | 118°53′ E | 31.32′ N | 157.58 | 14 18 | 0 |
| 7 | 118°54′ E | 31.39′ N | 187.93 | 15 18 | 0 |
| 8 | 118°65′ E | 31.42′ N | 169.46 | 16 20 | 1 |
| 9 | 118°67′ E | 31.54′ N | 123.93 | 16 19 | 1 |
| 10 | 118°57′ E | 31.58′ N | 147.58 | 17 22 | 0 |
| 11 | 118°37′ E | 31.62′ N | 151.16 | 17 24 | 0 |
| 12 | 118°65′ E | 31.64′ N | 163.51 | 18 22 | 0 |
| 13 | 118°83′ E | 31.79′ N | 174.62 | 18 24 | 0 |

In the dynamic operation scenarios: (1) Addition of new orders. It is assumed that new orders were added on 13 April, the earliest start time of the new order operation was April 14, the operation window duration of the new orders was in the range of 3–5 days, and the operation area of the new plant protection region is in the range of 100–200 hm$^2$. (2) Changes of transfer path. Influenced by COVID-19, traffic congestion, and other emergencies, some roads were under control on 17 April, and the distance between various plots was changed. It is assumed that the transfer distance between the plots is increased by 7 km in this paper. Detailed parameters of the new order and path are shown in Table 3.

**Table 3.** New order information.

| Number | Longitude | Latitude | Area/hm$^2$ | Operation Time Window | Urgency |
|--------|-----------|----------|-------------|------------------------|---------|
| 1 | 118°22′ E | 31°34′ N | 162.35 | 14–17 | 1 |
| 2 | 118°28′ E | 31°32′ N | 189.29 | 15–18 | 0 |
| 3 | 118°42′ E | 31°32′ N | 120.93 | 16–19 | 1 |
| 4 | 118°53′ E | 31°23′ N | 123.21 | 16–20 | 1 |
| 5 | 118°84′ E | 31°37′ N | 189.20 | 17–21 | 0 |
| 6 | 118°39′ E | 31°79′ N | 140.54 | 17–20 | 0 |

The parameters related to the operation and transfer of UAV plant protection teams are as follows. It is assumed that each plant protection UAV operates 8 h per day, the operation efficiency is 10 hm$^2$/h, and the transfer efficiency of the plant protection UAV is 30 km/h. In terms of operation cost and income, the operation income of the plant protection UAV is RMB 120/hm$^2$, the operation income of the rush order is RMB 140/hm$^2$, the operation energy cost is RMB 8/hm$^2$, the field operating machinery loss and pesticide cost is RMB 8/hm$^2$, and the transfer cost is RMB 0.90/km. The waiting cost of the plant protection UAV is RMB 1/h, and the cost of the operation delay is RMB 100/h.

### 5.2. Description of Simulation Environment

In this paper, the scheduling process of plant protection UAVs is simulated by computer. The computing device adopts a Windows 10 operating system with Intel i5 processor, and the simulation platform is MATLAB 2014b. To ensure full convergence of the model, the number of iterations of the simulated annealing algorithm and the Levi simulated annealing algorithm is 6000 times.

## 6. Results and Analysis

### 6.1. Initial Order Results and Analysis

In this paper, the Greedy Best First Search (GBFS), traditional simulated annealing (SA), and Levy simulated annealing (Levy-SA) algorithms are used to build the scheduling allocation models of plant protection UAVs. The process of the GBFS algorithm is as follows:

(1) In the order sorting sequence, the order at the top is taken.
(2) UAVs that are idle are identified, and the one closest to the current plot is selected to conduct the operation.
(3) Whether there are still orders in the order sorting sequence is determined. If so, go to (1); otherwise, end the process.

In this paper, the three scheduling methods are compared in five aspects: total operation income of the UAV plant protection teams, total scheduling distance, total operation time, total waiting time of the UAV plant protection teams, and operation delay time. The comparison effect is shown in Table 4.

**Table 4.** Original Order Scheduling Revenue.

| Modeling Method | Total Income/Yuan | Scheduling Distance/km | Operating Duration/h | Total Waiting Time/h | Delay Time/h |
|---|---|---|---|---|---|
| Greedy Best First Search algorithm (GBFS) | 18,323.35 | 330.50 | 72.46 | 343.00 | 0 |
| Simulated Annealing algorithm (SA) | 18,517.87 | 333.24 | 72.46 | 50.17 | 0 |
| Levy Annealing algorithm (Levy-SA) | 18,787.54 | 323.02 | 72.46 | 18.04 | 0 |

Table 4 indicates that all three scheduling methods can complete the operation within the required operation time window for the plot, and their total operation time is equal (72.46 h each). However, the total scheduling distance and waiting time of the UAV swarm based on the three methods are different: Both SA and Levy-SA outperform the GBFS in terms of scheduling distance and waiting time; in comparison, Levy-SA is superior to SA. The scheduling Gantt chart of Levy-SA is shown in Figure 6. The iterative process of Levy-SA and SA is shown in Figure 7. It can be seen that Levy-SA has a higher possibility of accepting the suboptimal solution than SA before 1000 iterations, and the function values fluctuate violently. After 1000 iterations, the probability of Levy-SA accepting the suboptimal solution decreases, and the objective function value tends to be stable and rises slowly. The function tends to converge after 3000 iterations. In comparison, SA shows a certain periodicity of iterative convergence in 6000 iterations, and it has a relatively high probability of accepting the suboptimal solution in the early or later stages of the iterations, which leads to the unstable function iteration effect and slow convergence.

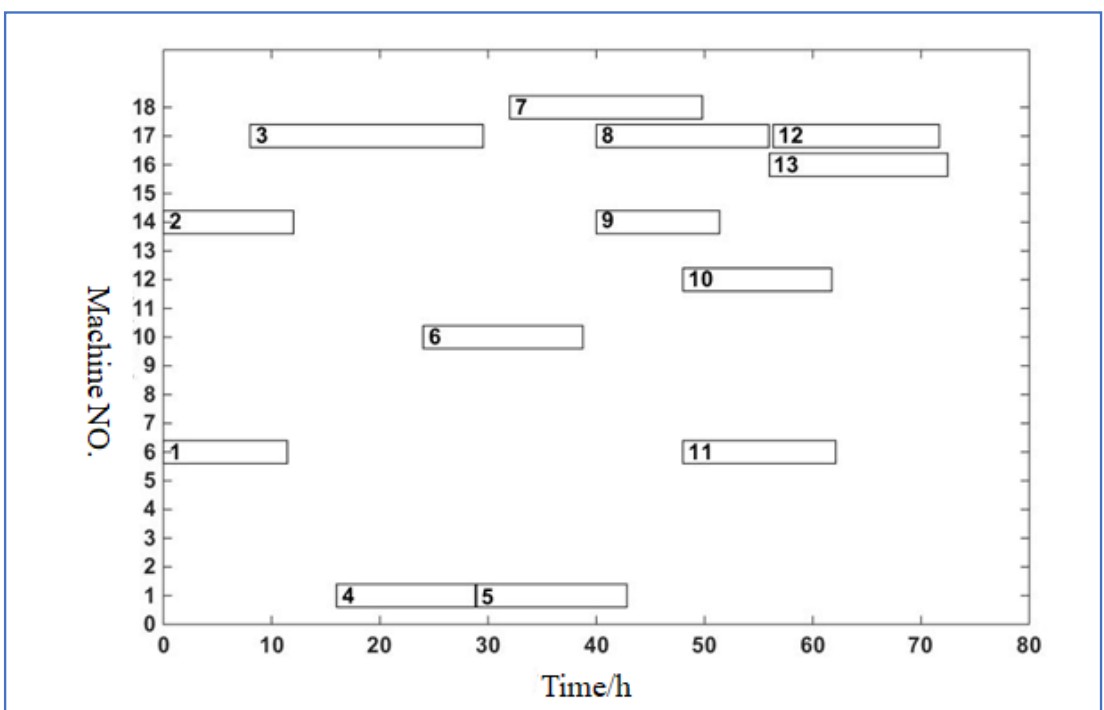

**Figure 6.** Scheduling Gantt chart of the original order by Levy-SA.

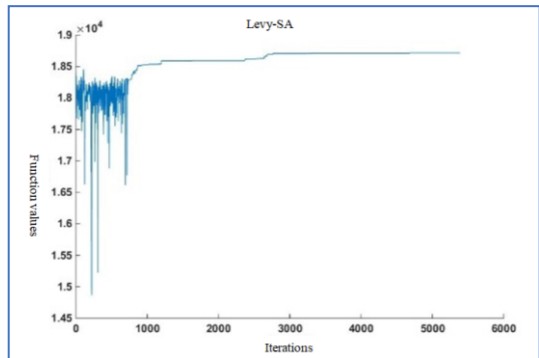
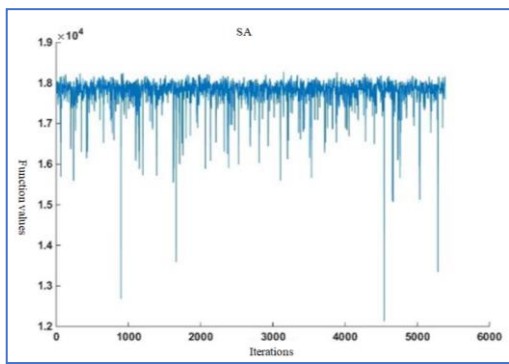

(**a**) Levy-SA Iteration Graph        (**b**) SA Iteration Graph

**Figure 7.** Iterative graph of Levy-SA and SA algorithms.

*6.2. New Order Results and Analysis*

According to the description of operation scenarios in Section 5.1, new orders are added after 13 days, and the orders are re-sorted and scheduled for assignment by the method proposed in Section 4.1. Firstly, the original unoperated orders at that moment are identified. The sorting weights for the original unoperated orders and new orders are calculated according to Equation (9), and the orders are sorted in descending order. On this basis, the order scheduling strategy is arranged by GBFS, SA, and Levy-SA. The indexes related to scheduling are shown in Table 5.

**Table 5.** Scheduling benefits after new orders are added.

| Method | Total Income/Yuan | Scheduling Distance/km | Operating Duration/h | Total Waiting Time/h | Delay Time/h |
|---|---|---|---|---|---|
| Greedy Best First Search algorithm (GBFS) | 20,639.97 | 630.08 | 72.46 | 525 | 0 |
| Simulated Annealing algorithm (SA) | 21,791.65 | 562.74 | 72.46 | 58.01 | 0 |
| Levy Annealing algorithm (Levy-SA) | 22,020.29 | 509.91 | 72.46 | 49.81 | 0 |

Table 5 indicates that the scheduling strategies of GBFS, SA, and Levy-SA are all equal in operation duration (72.46 h). The main reason is that the duration and the interval of operation time window required in the new order is essentially the same as in the original one, and the UAV plant protection team can still handle the current workload of order operation. There is no operation time delay based on any of the three algorithm, while SA and Levy-SA are superior to GBFS in the scheduling distance and waiting time of the strategies arranged. Both have advantages in scheduling distance and waiting time, while Levy-SA outperforms SA in the ability to jump out of the local optimal solution during the search process. The scheduling Gantt chart of Levy-SA for the original and new orders is shown in Figure 8. Through comparison with Figure 6, it can be seen that the original scheduling results are maintained for orders 1–4; original orders 5–13 and subsequent new orders are rescheduled according to the operation requirements and costs. The final goal of maximizing the operation income of the UAV plant protection teams is achieved.

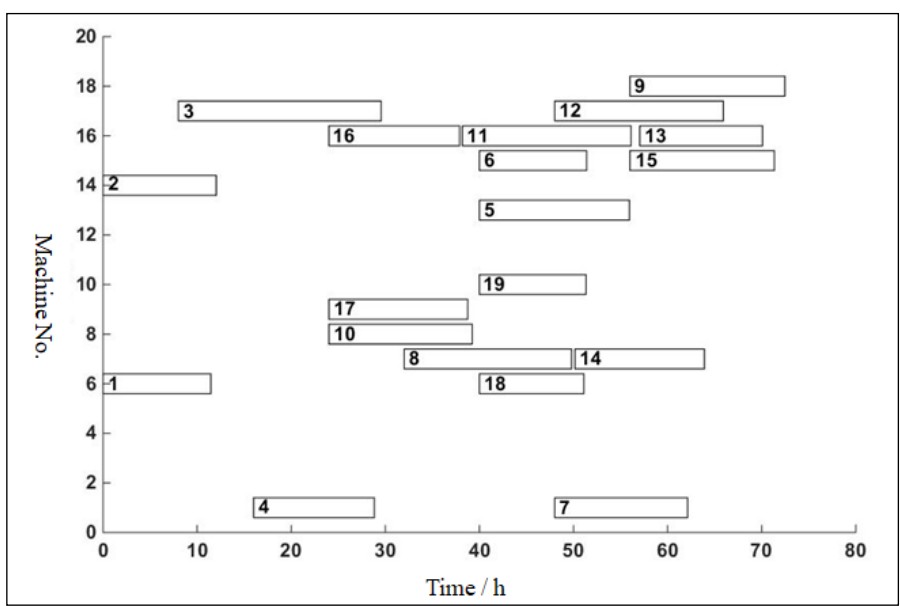

**Figure 8.** Scheduling Gantt Chart for New Orders by Levy-SA.

*6.3. Path Cost Change Results and Analysis*

According to the description of operation scenarios in Section 5.1, after 17 days, the path transfer cost between various fields is changed due to COVID-19 lockdown or road congestion. The orders are re-sorted and scheduled for allocation by the method proposed in Section 4.2. Firstly, the orders that are not operated at that moment are identified, based on which the order scheduling strategy is arranged by the methods in this paper. The scheduling results are shown in Table 6.

**Table 6.** Scheduling benefits after changing transfer cost.

| Method | Total Income/Yuan | Scheduling Distance/km | Operating Duration/h | Total Waiting Time/h | Delay Time/h |
|---|---|---|---|---|---|
| Original scheduling sequence | 21,789.16 | 709.91 | 72.46 | 49.81 | 0 |
| Greedy Best First Search algorithm (GBFS) | 21,856.23 | 580.73 | 75.62 | 49.81 | 0 |
| Simulated Annealing algorithm (SA) | 22,098.48 | 532.73 | 78.48 | 92.17 | 0 |
| Levy Annealing algorithm (Levy-SA) | 22,438.87 | 336.61 | 71.35 | 46.02 | 0 |

Table 6 indicates that the original scheduling strategy, GBFS, SA, and Levy-SA can all complete the plant protection operation within the specified time. In the total operation time consumption, both the original scheduling strategy and Levy-SA can complete the operation within 72.46 h, while it takes SA 78.48 h. In the scheduling distance, the process of the SA dynamic scheduling strategy saves 177.18 km and Levy-SA saves 373.3 km, as compared to the original scheduling method. In the waiting time, SA spends 42.36 h more and Levy-SA spends 3.79 h less than the original scheduling sequence. Figure 9 shows the scheduling Gantt chart of the dynamic scheduling strategy based on the methods in this paper when the field transfer cost of UAV plant protection teams is changed. The results indicate that compared to Figure 6, the scheduling arranged based on the dynamic

scheduling strategy results in a longer time interval between different tasks for each UAV after the path cost change. Moreover, plant protection tasks are assigned to each UAV based on the scheduling strategy with shorter scheduling distance, while the waiting time between different tasks is minimized for each UAV.

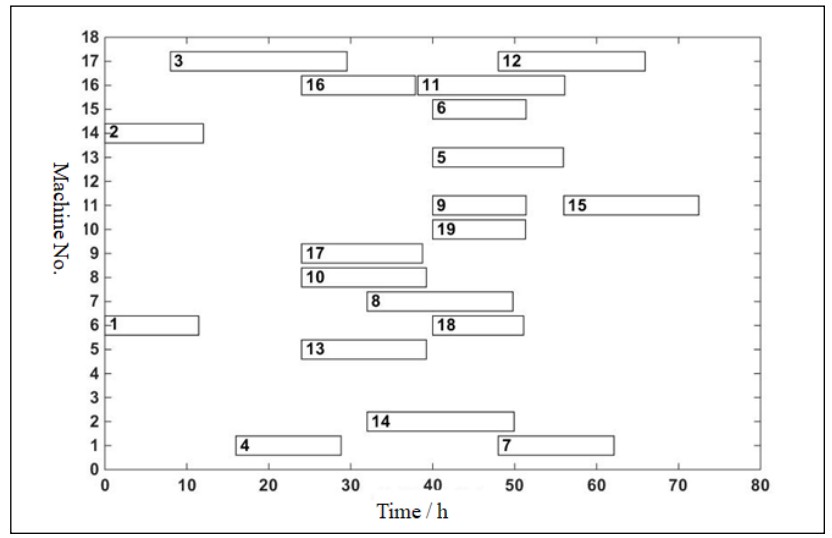

**Figure 9.** Scheduling Gantt chart after adjustment of field distance by Levy SA.

*6.4. Impact of Emergency Parameters on Scheduling Schemes*

Emergencies have an impact on flight defense and scheduling results, such as operation revenue, transfer mileage, waiting time, etc. The impact of different levels of emergencies on scheduling results will vary. To find out the changing trend of scheduling results with emergencies, based on the analysis of Section 6.2 results, different new order areas and time windows are introduced for repeated tests. Based on the data in Table 3, first fix the time window of the original new order and set the average area of the new order as 10 hm$^2$, 50 hm$^2$, 150 hm$^2$, and 200 hm$^2$. Then, fix the area of the original new orders and set the average time window of new orders to 4, 6, 8, and 10 d. Ten repeated tests were carried out with Levy annealing (Levy-SA), and the results are shown in Tables 7 and 8.

**Table 7.** The effect of the change of orders area on the scheduling result.

| Area/ hm$^2$ | Total Income/Yuan | | Scheduling Distance/km | | Operating Duration/h | | Total Waiting Time/h | | Delay Time/h | |
|---|---|---|---|---|---|---|---|---|---|---|
| | Mean Value | Standard Deviation | Mean Value | Standard Deviation | Mean Value | Standard Deviation | Mean Value | Standard Deviation | Mean Value | Standard Deviation |
| 10 | 18,607.46 | 107.48 | 546.05 | 53.74 | 72.46 | 0 | 49.58 | 45.59 | 0 | 0 |
| 50 | 21,382.39 | 101.41 | 563.59 | 50.71 | 72.46 | 0 | 47.62 | 13.92 | 0 | 0 |
| 150 | 27,720.77 | 700.51 | 574.14 | 17.67 | 72.46 | 0 | 51.45 | 7.09 | 0 | 0 |
| 200 | 30,211.63 | 90.43 | 598.66 | 45.09 | 72.46 | 0 | 72.24 | 5.86 | 0 | 0 |

**Table 8.** The effect of the change of operation time windows on the scheduling result.

| Operation Time/d | Total Income/Yuan | | Scheduling Distance/km | | Operating Duration/h | | Total Waiting Time/h | | Delay Time/h | |
|---|---|---|---|---|---|---|---|---|---|---|
| | Mean Value | Standard Deviation | Mean Value | Standard Deviation | Mean Value | Standard Deviation | Mean Value | Standard Deviation | Mean Value | Standard Deviation |
| 4 | 27,082.25 | 85.21 | 572.62 | 37.53 | 73.07 | 1.27 | 54.53 | 3.99 | 0 | 0 |
| 6 | 27,118.82 | 115.74 | 557.35 | 58.95 | 73.07 | 1.27 | 54.47 | 8.53 | 0 | 0 |
| 8 | 27,144.25 | 121.44 | 538.61 | 61.98 | 73.07 | 1.27 | 52.40 | 5.73 | 0 | 0 |
| 10 | 27,175.52 | 121.56 | 529.01 | 65.42 | 73.07 | 1.27 | 56.61 | 6.65 | 0 | 0 |

It can be seen from Table 7 that when the operation area of new orders increases, the total revenue of the UAV plant protection teams increases correspondingly, the total scheduling distance increases slightly, the total operation duration remains unchanged, and the delay time remains 0, which is basically consistent with the previous analysis structure. The total waiting time shows a trend of increasing first and then decreasing, and its standard deviation decreases with the increase in the work area. The main reason is that the smaller the area of new orders, the smaller the proportion of the actual work time in the time window, which will lead to large fluctuations in the optimization results. In actual production, independent small orders will also increase production costs, and the flight defense team's handling schemes for such orders are mostly denial of service or increase of service unit price, which is basically consistent with the model analysis results.

It can be seen from Table 8 that the time window of new orders is enlarged, the total revenue of the UAV plant protection teams and the total waiting time have not changed much, and the total delay time remains zero. The total operation time remained unchanged, but slightly increased by 0.61 h compared with the results in Table 7, and the standard deviation changed from 0 to 1.27, mainly due to 72.46 h for 8 out of 10 simulation test results, but 75.07 h for 2 of them. According to the test results reflected in Tables 7 and 8, the dynamic model designed in this paper has good adaptability to different levels of emergencies, but it is necessary to pay special attention to the fact that orders with too small size may cause certain interference to the results.

## 7. Discussion

In this paper, innovations are made in two aspects: scheduling model solving methods and dynamic scenarios during the operation of UAV plant protection teams, which are discussed as follows.

In the scheduling model solving method, SA is improved by the Levy distribution method in this paper. The Levy distribution probability function has the following characteristics: the function value is small on large probability and large on small probability. When applied in the probability function of SA for accepting suboptimal solutions, the probability of accepting suboptimal solutions is relatively high when the temperature is high, while the probability of accepting suboptimal solutions gradually decreases as the iteration increases and the temperature keeps dropping. In the scheduling results of the scheduling model, Levy-SA outperforms SA and GBFS in the total operation income, total operation time, waiting time, transfer path, and other aspects. From previous studies [34,35], all heuristic algorithms improved by the Levy distribution method have higher search ability in the early stage and faster convergence in the later stage of search than the original algorithm. In the search results, all heuristic algorithms improved by Levy distribution can obtain better search results.

In the establishment of the dynamic scheduling model, the traditional scheduling model generally addresses the static situation, where the orders of plant protection and the location of plant protection UAVs are known [11–13]. However, unexpected situations occur frequently in the practical operation process, such as the addition of new orders and changes in the transfer cost of the UAV plant protection teams [7]. These emergencies may have a significant or even subversive impact on the established initial production plan [40]. If the study only solves the UAV scheduling scheme through a simple static optimization model and ignores the impact of emergencies, the scheduling results obtained will only have a certain reference significance and greatly reduce the application value in actual production [41,42]. In this paper, orders change and traffic change were introduced to establish a dynamic scheduling model to improve the practicability of the model. This paper draws on the idea of dynamic scheduling models in logistics and military fields and transforms dynamic events into a static scheduling model with event-driven search, with unexpected events as the driving events of the scheduling model. The scheduling results indicate that the scheduling strategy developed by the dynamic scheduling model has higher economic efficiency and less time consumption than the scheduling model

developed in the static situation, and it is better adapted to the practical situation when unexpected events occur.

This paper also has some limitations. The emergencies introduced in the dynamic model are hypothetical events in advance, which have a limited impact on the global scheduling scheme. However, in reality, some emergencies are completely unpredictable and uncertain events, such as major weather changes that interfere with the normal operation of the UAV [43,44]. Major traffic accidents lead to traffic congestion in some sections for a certain period. The migration of pest groups reverses the risk level of diseases and pests in different regions. Such interference events will have a subversive impact on the dynamic scheduling scheme. Aircraft fault, battery endurance, pharmaceutical capacity, and other aircraft operating conditions may also change, which will affect the scheduling scheme. Due to the lack of such monitoring data, this paper does not involve the analysis of the impact of aircraft operating conditions on the scheduling scheme. Therefore, in the follow-up study, the dynamic scheduling model will be improved by collecting data related to the weather forecast, pest forecast, traffic forecast, etc., as well as collecting aircraft working conditions data through sensors for the joint analysis of big data mining and dynamic planning, to improve the practicability of the scheduling scheme.

## 8. Conclusions

(1) To address the problems that SA is prone to falling into the local optimal solution in the early stage and converges slowly in the later stage, it is proposed that SA is optimized by the Levy distribution function to form Levy-SA.

(2) An order allocation model of UAV plant protection is designed, focusing on the dynamic allocation model for new orders and transfer path changes during the operation process. The allocation strategy is optimized by Levy-SA. The results indicate that the optimization strategy of Levy-SA is superior to the SA and GBFS in the total operation income, operation time, scheduling distance, and waiting time of UAV plant protection teams.

**Author Contributions:** Conceptualization, C.C. and G.C.; methodology, C.C. and Y.L.; software, Y.L.; validation, C.C. and J.Z.; formal analysis, C.C. and Y.L.; investigation, G.C.; resources, J.Z.; data curation, Y.L.; writing—original draft preparation, C.C.; writing—review and editing, C.C. and Y.L.; visualization, C.C.; supervision, C.C.; project administration, G.C.; funding acquisition, G.C. All authors have read and agreed to the published version of the manuscript.

**Funding:** This research was funded by a grant of the special funding for basic scientific research business expenses of central public welfare scientific research institutes (S202215).

**Institutional Review Board Statement:** Not applicable.

**Informed Consent Statement:** Not applicable.

**Data Availability Statement:** Data are available from the first author on demand.

**Acknowledgments:** All the authors are very grateful to the anonymous reviewers and editors for their careful review and critical comments.

**Conflicts of Interest:** The authors declare no conflict of interest.

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
