# Peer review of "Research on Dynamic Scheduling Model of Plant Protection UAV Based on Levy Simulated Annealing Algorithm"

_sustainability, doi:10.3390/su15031772_

Round 1

Reviewer 1 Report

Dear authors,

I have reviewed your manuscript " Research on Dynamic Scheduling Model of Plant Protection UAV Based on Levy Simulated Annealing Algorithm " submitted for publication in Sustainability. The manuscript is interesting and presents a valuable collection of information on the use of Levi distribution method to improve the simulated annealing algorithm, and the results demonstrated that the proposed method could meet the needs of plant protection UAV scheduling operations in static and dynamic scenarios. Additionally, this manuscript provides provide a reference for the development of agricultural machinery intelligent scheduling system. However, I have highlighted a few points below that could improve the quality of the manuscript before publication. The discussion needs to be improved; in the current version there are only two references cited.

Line 7: Please double check if the email address is correct 751746559@qq .com.

Line 8: Please describe “UAV”

Line 9: “UAV plant protection team” Why is this so specific? this could benefit other teams as well.

Line 10: simulated annealing algorithm (SA).

Line 11: Please use the abbreviation SA

Lines 21-22: that the traditional SA is easy…

Line 28: Same observation as above; you can use the abbreviation

Line 65-66: “started earlier”. Please consider adding the year or decade [add a reference to support this sentence].

Line 81: [20-22]. The exact           (you forgot to add the punctuation after the references)

Line 85: problems [add a reference to support this sentence].

Line 85: simulated annealing algorithm (SA)

Line 86: algorithm [add a reference to support this sentence].

Line 95: include: operation

Line 96: The above studies.. Which studies? Please refer them.

Line 105: it seems that there is a double space between scheduling and studies..

Line 106: types: prediction

Line 108: events include those from Amorim et al. [32] and Chang et al. [33].

Line 114: Same observation as above “Studies on event-driven methods include those from xx and xx

Lines 350-351: references [20-22],

Please describe the abbreviations in the figures and tables; for example “SA”, “GBFS”, etc.

Line 409: use “h” instead of “hours”. Please standardize this throughout the document.

Line 454: Increase font size on chart axes.

Line 457: Increase font size on chart axes.

Lines 482 and 506: Same observation as above.

Line 507: Discussion is too short. Consider adding more references to your discussion; in the current version there are only two references cited.

Line 520: “[31-32]” should not be superscript.

Line 559: It is necessary to format the references according to the requirements of the journal. Please check this information here https://www.mdpi.com/authors/references

Author Response

Thank you for your valuable comments. I have revised the manuscript according to your comments, please see the attachment for details

Reviewer 2 Report

This paper proposes to use the Levi distribution method to improve the simulated annealing algorithm, and proposes a dynamic scheduling model driven by unexpected events such as new orders and transfer path changes. But there are still some contents, which need be revised in order to meet the requirements of publish. A number of concerns listed as follows:

(1) In the introduction, the authors should clearly indicate the contributions and innovations of this paper.

(2) Please highlight your contributions in introduction.

(3) The computation complexity of the proposed method should be clearly described.

(4) More statistical methods are recommended to analyze the experimental results.

(5) Analysis is insufficient. An extensive analysis is required.

(6) In Line 199, for the expression (6), these parameters should give and explain.

(7) Line 457, Figure 6 is not clear, please revise it to be clear.

(8) Literature survey is insufficient. You must add and review all significant similar works that have been done. For example, https://doi.org/10.3389/fendo.2022.1057089;

https://doi.org/10.1016/j.ymssp.2022.109422;

https://doi.org/10.3934/mbe.2023090;

https://doi.org/10.1109/TSMC.2020.3030792 and so on.

(9) Check carefully for a few clerical errors and formatting issues.

Author Response

(The authors gave the same response as above.)

Reviewer 3 Report

Well written. Language can be emphasized on plant protection more than engineering.

Author Response

Thank you very much for your recognition of the manuscript. The purpose of this paper is to improve the effect of emergency prevention and control of diseases and pests through the efficient use of unmanned aircraft, so it is to improve plant protection through engineering.

Round 2

Reviewer 2 Report

OK